# S2HPruner: Soft-to-Hard Distillation Bridges the Discretization Gap in Pruning

**Weihao Lin**[1][†] , **Shengji Tang**[1][†] , **Chong Yu**[2] , **Peng Ye**[3] , **Tao Chen**[1][*]

[1]School of Information Science and Technology, Fudan University, Shanghai, China,
[2]Academy for Engineering and Technology, Fudan University, Shanghai, China,
[3]Shanghai AI Laboratory, Shanghai, China

`eetchen@fudan.edu.cn`

## Abstract

Recently, differentiable mask pruning methods optimize the continuous relaxation architecture (soft network) as the proxy of the pruned discrete network (hard network) for superior sub-architecture search. However, due to the agnostic impact of the discretization process, the hard network struggles with the equivalent representational capacity as the soft network, namely discretization gap, which severely spoils the pruning performance. In this paper, we first investigate the discretization gap and propose a novel structural differentiable mask pruning framework named S2HPruner to bridge the discretization gap in a one-stage manner. In the training procedure, S2HPruner forwards both the soft network and its corresponding hard network, then distills the hard network under the supervision of the soft network. To optimize the mask and prevent performance degradation, we propose a decoupled bidirectional knowledge distillation. It blocks the weight updating from the hard to the soft network while maintaining the gradient corresponding to the mask. Compared with existing pruning arts, S2HPruner achieves surpassing pruning performance without fine-tuning on comprehensive benchmarks, including CIFAR-100, Tiny ImageNet, and ImageNet with a variety of network architectures. Besides, investigation and analysis experiments explain the effectiveness of S2HPruner. Codes are publicly available on GitHub: `https://github.com/opposj/S2HPruner`.

## 1 Introduction

As deep neural networks (DNN) have achieved success in substantial fields [20, 58, 35, 61, 49], the increasing computation and storage cost of DNN impedes practical implementation. Model pruning [39, 57, 62], which aims at removing the less informative in a cumbersome network, has been a widespread technique for model compression. Pioneer pruning methods utilize regularization terms [63, 43] to sparsify the network or introduce importance metrics [30, 19, 18] to remove less important weights directly. However, due to the latent correlations between weights, simply eliminating the weights in an over-parameter model will hinder the integrality of structure, especially in structural pruning, where grouped filters are removed.

Recently, it has been pointed out that the structure of the pruned network is essential for the final pruning performance [40]. Inspired by the differentiable architecture search (DARTS) [36, 67, 7], emerging works [15, 17, 16], namely differentiable mask pruning (DMP), introduce learnable parameters to generate the weight mask and impose the task-aware gradient to guide the structure search of the pruned network. In the training procedure, DMP introduces the learnable mask into the gradient graph by coupling the mask with the activation feature or weights, e.g., directly multiplying

---

[*]Corresponding Author (eetchen@fudan.edu.cn).  [†]Equal Contribution.

38th Conference on Neural Information Processing Systems (NeurIPS 2024).

Figure 1: Comparison of different typical pruning methods and illustration of discretization gap. The darker color represents the higher relative magnitude scale of weights or masks. $\odot$ denotes Hadamard product. For ease of demonstration, we use one layer to represent the entire network.

the mask with the feature or weights. Through gradient descent, DMP can jointly optimize the weights and mask parameters for a bespoke structure and parameter distribution, thus causing a better performance. The search procedure essentially regards the mask-coupled network (soft network) as the performance proxy of the final discretized compact pruned network (hard network). Whereas, considering the aim of pruning is to obtain a capable hard network, a natural question is **whether a superior soft network implies a corresponding high-performance hard network**.

In DARTS, there is a problem known as the discretization gap [66, 60, 7], which refers to the discrepancy between the continuous relaxation architecture and the discrete architecture due to the discretization process. Since DMP follows a similar modeling format to DARTS, it also faces a comparable discretization gap problem[†] that the hard network struggles from having the semblable representational capacity as the soft network. A specific manifestation is that the hard network performs significantly poorer in the evaluation metrics than the soft network. Fig. 1 visually exhibits the different pruning methods and discretization gap. The discretization gap severely impacts pruning performance but has been long overlooked in DMP. There are potential techniques that may alleviate the discretization gap in previous works, e.g., gradually facilitating the steepness of the Sigmoid function via decaying temperature [26, 27, 44, 50] and optimizing the binary mask via the straight-through estimator (STE) [65, 15]. However, these methods lead to certain side effects: the decaying temperature results in difficult mask optimization because of the vanishing gradient, and STE causes a suboptimal mask due to the coarse gradient.

To alleviate the discretization gap in DMP without influencing mask optimization, we formulate the mask pruning in a soft-to-hard paradigm and propose a structured differentiable mask pruning framework named Soft-to-Hard Pruner (S2HPruner). Specifically, in the training procedure, we not only forward the soft network for the structural search but also forward the corresponding hard network and distill it under the supervision of the soft network to reduce the discretization gap. Meanwhile, we discover that even with the same corresponding hard network, the distribution of the mask parameters influences the discretization gap essentially. However, the common unidirectional knowledge distillation (KD) cannot optimize mask parameters directly, but bidirectional KD causes unbearable performance degradation. Therefore, we propose a decoupled bidirectional KD, which blocks the weight updating from the hard to the soft network while keeping the gradient corresponding to the mask. Exhaustive experiments on three mainstream classification datasets, including CIFAR-100, Tiny ImageNet, and ImageNet, demonstrate the effectiveness of S2HPruner.

Our contributions are summarised as follows:

- We first study and reveal the long-standing overlooked discretization gap problem in differentiable mask pruning. To alleviate it, we propose a soft-to-hard distillation paradigm, which distills the hard network under the supervision of the soft network.

- Based on the soft-to-hard knowledge distillation paradigm, we propose a novel differentiable mask pruning framework named Soft-to-Hard Pruner (S2HPruner). To further reduce the

---

[†]To avoid confusion, the discretization gap discussed following is in the context of DMP.

discretization gap and avoid performance degradation, we propose a decoupled bidirectional KD which blocks and allows the gradient of model weights and mask parameters selectively.

- Extensive experiments on three mainstream datasets and five architectures verify the superiority of S2HPruner, e.g., maintaining 96.17%(Top-1 accuracy 73.23% in 76.15%) with around 15% FLOPs. Additional ablation and investigation experiments demonstrate the underlying mechanism of the effectiveness.

## 2 Related works

### 2.1 Differentiable mask pruning

Considering the network structure has a decisive impact on the pruning performance [40], numerous works [15, 12] train a binary mask for an optimal selection of sub-architecture. However, because of the non-differentiable property, directly optimizing the binary mask is very challenging and even impairs the performance [26]. Differently, differentiable mask pruning (DMP) methods [17, 9, 4, 27, 44] adopt differentiable continuous relaxation as a performance proxy of the hard network for structure search, which can be easily optimized by task-aware loss end-to-end. DMCP [17] regards the channel pruning as a Markov process and builds a differentiable mask based on the transitions between states. AutoPruner [44] proposes to construct a meta-network to generate the differentiable mask according to the activation responses, and a scaled temperature facilitates the sigmoid function approaching step function to obtain an approximate binary mask. GAL [34] learns a differentiable mask by optimizing a generative adversarial learning task in a label-free and end-to-end manner. However, the task-aware loss can ensure the high performance of the soft network but not the final hard network. There is a discretization gap limiting the target hard network during the discretization process. Different from previous DMP methods, which only focus on optimizing the soft network, our approach aims to achieve a high-performance hard network by reducing the discretization gap through soft-to-hard distillation.

### 2.2 Pruning with distillation

As a network compression technique orthogonal to pruning, knowledge distillation [22, 28, 52] (KD) transfers the dark knowledge from a large teacher network to enhance a compact student network. Recently, there have been substantial works [46, 47, 3, 32, 10] introducing KD into model pruning to further boost the pruned network. JMC [10] proposes a structured pruning based on the magnitude of weights and a many-to-one layer mapping strategy to distill the dense model to the pruned one. KD ticket [46] exploits the dark knowledge in the early stage of iterative magnitude pruning to boost the lottery tickets in the dense model. DIPNet [72] improves the ability of the pruned model by the supervision of high-resolution output. The above methods treat KD as an independent plug-in technique to enhance pruning performance without tight coupling with the selection of weights. Differently, in the proposed method, KD contributes to mask optimization directly as an integral part of the core pruning procedure. Moreover, in contrast to the typical unidirectional KD, we propose a novel decoupled bidirectional KD to alleviate the discretization gap between soft and hard networks, due to the distinct attributes of mask and weights.

## 3 Method

### 3.1 Problem formulation

Given a network with parameters $\boldsymbol{\theta}$, a pruning algorithm generates a binary mask $\boldsymbol{m}$ via solving the following constraint optimization:

$$\min_{\boldsymbol{\theta}, \boldsymbol{m}} \mathcal{L}\left(\boldsymbol{\theta}\left\langle\boldsymbol{m}\right\rangle\right) \quad \text{s.t. } \mathcal{R}\left(\boldsymbol{m}, T\right) = 0. \tag{1}$$

The $\boldsymbol{\theta}\left\langle\boldsymbol{m}\right\rangle$ are the remaining parameters after pruning. The $\mathcal{L}$ and $\mathcal{R}$ are the task-specific performance loss and resource regularization, respectively. The $T$ is a manually assigned resource budget. Intuitively, a pruning algorithm attains a slimmed subnet that optimally balances the performance and the resource consumption.

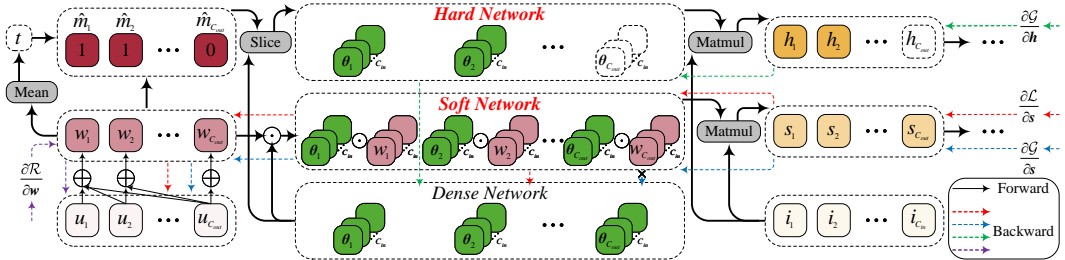

Figure 2: The proposed pruner's forward and backward flows, illustrated via an exemplary linear layer with parameters $\boldsymbol{\theta}$. The $\boldsymbol{u}$ are the additional learnable parameters normalized by softmax. The $\boldsymbol{w}$ denotes the relaxed mask. The estimated binary pruning mask is the $\hat{\boldsymbol{m}}$. The input is denoted by $\boldsymbol{i}$. The output of the soft and hard networks are the $\boldsymbol{s}$ and $\boldsymbol{h}$, respectively. The $\mathcal{L}$, $\mathcal{G}$, and $\mathcal{R}$ are the performance loss, gap measure, and resource regularization, respectively.

## 3.2 Overview

Directly optimizing the problem 1 is almost intractable due to the discreteness of $\boldsymbol{m}$. To get around, we introduce a relaxation of $\boldsymbol{m}$ as $\boldsymbol{w}$, which is continuous and bounded to $[0, 1]$. The $i$-th element in $\boldsymbol{w}$ represents the probability of the $i$-th parameter being retained. Consequently, a differentiable representative for $\boldsymbol{\theta}\langle\boldsymbol{m}\rangle$ can be constructed as $\boldsymbol{\theta}\odot\boldsymbol{w}$, where the $\odot$ denotes the Hadamard product. Based on this relaxation, the problem 1 can be reformulated as two parts:

$$\text{Part 1: } \min_{\boldsymbol{\theta},\boldsymbol{w}}\left(\mathcal{L}\left(\boldsymbol{\theta}\odot\boldsymbol{w}\right)+\alpha\mathcal{R}\left(\boldsymbol{w},T\right)\right),$$
$$\text{Part 2: } \min_{\boldsymbol{\theta},\boldsymbol{w}}\mathcal{G}\left(\boldsymbol{\theta}\langle\hat{\boldsymbol{m}}\rangle,\boldsymbol{\theta}\odot\boldsymbol{w}\right). \tag{2}$$

The $\alpha$ is a Lagrangian multiplier, regarded as a hyperparameter. The $\mathcal{G}$ is a gap measure, reflecting the difference between $\boldsymbol{\theta}\langle\hat{\boldsymbol{m}}\rangle$ and $\boldsymbol{\theta}\odot\boldsymbol{w}$. The $\hat{\boldsymbol{m}}$ is an estimated pruning mask, derived from $\boldsymbol{w}$ as $\mathbb{I}_{[t,1]}\left(\boldsymbol{w}\right)$, where the $\mathbb{I}$ is an indicator function, and the t is a threshold. In the problem 2, the first part searches for a high-performance soft network that satisfies the resource constraint, and the second part reduces the gap between the hard network and the soft one. Similar to [36, 33], to avoid alternate optimization, we combine the two parts with two additional hyperparameters $\beta$ and $\gamma$:

$$\min_{\boldsymbol{\theta},\boldsymbol{w}}\left(\beta\mathcal{L}\left(\boldsymbol{\theta}\odot\boldsymbol{w}\right)+\alpha\beta\mathcal{R}\left(\boldsymbol{w},T\right)+\gamma\mathcal{G}\left(\boldsymbol{\theta}\langle\hat{\boldsymbol{m}}\rangle,\boldsymbol{\theta}\odot\boldsymbol{w}\right)\right). \tag{3}$$

The problem 3 is differentiable w.r.t. both $\boldsymbol{\theta}$ and $\boldsymbol{w}$, thus can be optimized by gradient-based methods [41, 54]:

$$\Delta\boldsymbol{\theta}=-\lambda_{\boldsymbol{\theta}}\left(\beta\boldsymbol{g}_{\mathcal{L}\to\boldsymbol{\theta}\odot\boldsymbol{w}\to\boldsymbol{\theta}}+\gamma\boldsymbol{g}_{\mathcal{G}\to\boldsymbol{\theta}\langle\hat{\boldsymbol{m}}\rangle\to\boldsymbol{\theta}}+\gamma\boldsymbol{g}_{\mathcal{G}\to\boldsymbol{\theta}\odot\boldsymbol{w}\to\boldsymbol{\theta}}\right),$$
$$\Delta\boldsymbol{w}=-\lambda_{\boldsymbol{w}}\left(\beta\boldsymbol{g}_{\mathcal{L}\to\boldsymbol{\theta}\odot\boldsymbol{w}\to\boldsymbol{w}}+\alpha\beta\boldsymbol{g}_{\mathcal{R}\to\boldsymbol{w}}+\gamma\boldsymbol{g}_{\mathcal{G}\to\boldsymbol{\theta}\odot\boldsymbol{w}\to\boldsymbol{w}}\right). \tag{4}$$

The $\lambda_{\boldsymbol{\theta}}$ and $\lambda_{\boldsymbol{w}}$ are learning rates for $\boldsymbol{\theta}$ and $\boldsymbol{w}$, respectively. The $\boldsymbol{g}_X$ denotes the gradient obtained via a backward path $X$. Note that the term $\boldsymbol{g}_{\mathcal{G}\to\boldsymbol{\theta}\odot\boldsymbol{w}\to\boldsymbol{\theta}}$ implies aligning the soft network towards the hard one, which would severely deteriorate the performance of the soft network (see Section 4.2 for details). Consequently, the update of $\boldsymbol{\theta}$ is modified to:

$$\Delta\boldsymbol{\theta}=-\lambda_{\boldsymbol{\theta}}\left(\beta\boldsymbol{g}_{\mathcal{L}\to\boldsymbol{\theta}\odot\boldsymbol{w}\to\boldsymbol{\theta}}+\gamma\boldsymbol{g}_{\mathcal{G}\to\boldsymbol{\theta}\langle\hat{\boldsymbol{m}}\rangle\to\boldsymbol{\theta}}\right). \tag{5}$$

The essence of the above optimization lies in two aspects: 1) the joint optimization of the entire parameters $\boldsymbol{\theta}\odot\boldsymbol{w}$ and a dynamic subset of parameters $\boldsymbol{\theta}\langle\hat{\boldsymbol{m}}\rangle$ benefits from stimulative training [68], where the entire parameters transfer knowledge to the partial ones, and the improvement of the partial parameters can, in turn, enhance the entire ones; 2) the optimization of $\boldsymbol{w}$ involves the soft-to-hard gap, which provides a new dimension to bridge the gap besides adjusting the parameters. The pseudo-code describing the whole training process can be referred to in Algorithm 1, and a visualization of the forward/backward passes is provided in Fig. 2.

**Algorithm 1:** The training pseudo-code based on Pytorch automatic differentiation

---

**Input:** Initialized $\boldsymbol{\theta}^0$ and $\boldsymbol{w}^0$, iteration limit $i_{max}$, dataset $\mathcal{D}$, network forward function $\mathcal{N}$, resource budget $T$, performance metric $\mathcal{L}$, resource regularization $\mathcal{R}$, gap measure $\mathcal{G}$, pruning threshold $t$, gradient-based optimizer $\mathcal{O}$, hyperparameters $\alpha$, $\beta$, and $\gamma$

**Output:** $\boldsymbol{\theta}^{i_{max}}$ and $\hat{\boldsymbol{m}}^{i_{max}} = \mathbb{I}_{[t,1]}\left(\boldsymbol{w}^{i_{max}}\right)$

1   $i \leftarrow 0$;
2   **while** $i < i_{max}$ **do**
3     Fetch a sample $\boldsymbol{x}$ with its label $\boldsymbol{y}$ from $\mathcal{D}$;
4     $\boldsymbol{y}_s \leftarrow \mathcal{N}\left(\boldsymbol{\theta}^i \odot \boldsymbol{w}^i\right)$;                 `// The forward pass of the soft network`
5     $\boldsymbol{y}_h \leftarrow \mathcal{N}\left(\boldsymbol{\theta}^i \left\langle \mathbb{I}_{[t,1]}\left(\boldsymbol{w}^i\right)\right\rangle\right)$;        `// The forward pass of the hard network`
6     $l \leftarrow \mathcal{L}\left(\boldsymbol{y}_s, \boldsymbol{y}\right)$; $r \leftarrow \mathcal{R}\left(\boldsymbol{w}^i, T\right)$;
7     $d_1 \leftarrow \mathcal{G}\left(\boldsymbol{y}_h, \boldsymbol{y}_s.\,\mathrm{detach}\,()\right)$; $d_2 \leftarrow \mathcal{G}\left(\boldsymbol{y}_h.\,\mathrm{detach}\,(), \boldsymbol{y}_s\right)$;
8     $\left(\boldsymbol{g}_{\mathcal{L}\to\boldsymbol{\theta}\odot\boldsymbol{w}\to\boldsymbol{\theta}}, \boldsymbol{g}_{\mathcal{L}\to\boldsymbol{\theta}\odot\boldsymbol{w}\to\boldsymbol{w}}, \boldsymbol{g}_{\mathcal{R}\to\boldsymbol{w}}\right) \leftarrow (l + r).\,\mathrm{backward}\,()$;
9     $\boldsymbol{g}_{\mathcal{G}\to\boldsymbol{\theta}\langle\hat{\boldsymbol{m}}\rangle\to\boldsymbol{\theta}} \leftarrow d_1.\,\mathrm{backward}\,()$;
10    $\boldsymbol{g}_{\mathcal{G}\to\boldsymbol{\theta}\odot\boldsymbol{w}\to\boldsymbol{w}} \leftarrow d_2.\,\mathrm{backward}\,\left(\mathrm{inputs} = \boldsymbol{w}^i\right)$;
11    $\boldsymbol{\theta}^{i+1} \leftarrow \mathcal{O}\left(i, \boldsymbol{\theta}^i, \beta\boldsymbol{g}_{\mathcal{L}\to\boldsymbol{\theta}\odot\boldsymbol{w}\to\boldsymbol{\theta}} + \gamma\boldsymbol{g}_{\mathcal{G}\to\boldsymbol{\theta}\langle\hat{\boldsymbol{m}}\rangle\to\boldsymbol{\theta}}\right)$;      `// Eq. 5`
12    $\boldsymbol{w}^{i+1} \leftarrow \mathcal{O}\left(i, \boldsymbol{w}^i, \beta\boldsymbol{g}_{\mathcal{L}\to\boldsymbol{\theta}\odot\boldsymbol{w}\to\boldsymbol{w}} + \alpha\beta\boldsymbol{g}_{\mathcal{R}\to\boldsymbol{w}} + \gamma\boldsymbol{g}_{\mathcal{G}\to\boldsymbol{\theta}\odot\boldsymbol{w}\to\boldsymbol{w}}\right)$;    `// Eq. 4`
13    $i \leftarrow i + 1$

---

### 3.3 Implementation details

We focus on dependency-group-based structural pruning [6, 14], where layers in the same group share a single mask and are pruned as a whole. Besides, the pruning mask is channel-wise to comply with the structural pattern. The performance metric $\mathcal{L}$ is the cross-entropy for classification. The Kullback-Leibler divergence is selected as the gap measure $\mathcal{G}$.

**Acquisition of $\boldsymbol{w}$ and $t$**    Consider a linear layer parameterized by $\boldsymbol{\theta} \in \mathbb{R}^{C_{out} \times C_{in}}$. The corresponding binary pruning mask is denoted as $\boldsymbol{m} \in \mathbb{B}^{C_{out}}$. To generate $\boldsymbol{w}$, we define learnable parameters $\boldsymbol{u} \in \mathbb{R}^{C_{out}}$, which can be normalized to $[0, 1]$ via a softmax function. After softmax, the $i$-th element in $\boldsymbol{u}$ can be interpreted as the probability of retaining the first $i$ channels. Consequently, the probability of the $i$-th channel being retained, *i.e.*, $w_i$, can be calculated as $\sum_{k=i}^{C_{out}} u_k$. With the $\boldsymbol{w}$ obtained, the pruning threshold $t$ is derived as $\frac{1}{C_{out}} \sum_{k=1}^{C_{out}} w_k$.

**Resource regularization**    We utilize floating-point operations per second (FLOPs) to evaluate resource consumption. Given a target $T$ (in percentage), the resource regularization $\mathcal{R}$ is defined as $\left(\mathrm{FP}_{soft} / \mathrm{FP}_{all} - T\right)^2$. The $\mathrm{FP}_{all}$ is the FLOPs of the entire network. The $\mathrm{FP}_{soft}$ is the summation of layer-wise differentiable FLOPs. To be differentiable, the output channel number of a layer is calculated as $\sum_{k=1}^{C_{out}} \left(u_k * k\right)$. The $u_k$ is a softmaxed parameter introduced in the previous section.

## 4 Experiments

In this section, we begin by validating the effectiveness of the proposed pruner using three benchmark datasets: CIFAR-100 [29], Tiny ImageNet [11], and ImageNet [11]. For CIFAR-100 and Tiny ImageNet, we evaluate three common CNN architectures, *i.e.*, ResNet-50 [20], MobileNetV3 (MBV3) [24], and WRN28-10 [73], and two Transformer architectures, *i.e.*, ViT [61] and Swin Transformer [37], across various pruning ratios including 15%, 35%, and 55%. For ImageNet, ResNet-50 serves as the backbone model, and we compare the proposed pruner with several structural pruning methods in terms of Top-1 accuracy and FLOPs. After the benchmarking, investigative experiments are performed on CIFAR-100 using ResNet-50 to elucidate the influence of each gradient term in Algorithm 1 and the gap-narrowing capacity of the proposed pruner. Detailed training configurations are provided in the Appendix.

Table 1: The comparison of different pruning methods on CIFAR-100. We report the Top-1 accuracy(%) of dense and pruned networks with different remaining FLOPs.

| Method | ResNet-50 (Acc: 78.14) | | | MBV3 (Acc: 78.09) | | | WRN28-10 (Acc: 82.17) | | |
|---|---|---|---|---|---|---|---|---|---|
| | 15% | 35% | 55% | 15% | 35% | 55% | 15% | 35% | 55% |
| RST-S [1] | 75.02 | 76.38 | 76.48 | 72.90 | 76.78 | 77.30 | 78.56 | 81.18 | 82.19 |
| Group-SL [14] | 49.04 | 77.90 | 78.37 | 1.43 | 4.90 | 26.24 | 42.41 | 67.71 | 79.59 |
| OTOv2 [6] | 77.04 | 77.65 | 78.35 | 76.29 | 77.35 | 78.39 | 77.26 | 80.61 | 80.84 |
| Refill [5] | 75.12 | 77.43 | 78.19 | 69.57 | 75.91 | 76.96 | 75.98 | 79.25 | 79.56 |
| Ours | **79.77** | **79.87** | **80.10** | **77.28** | **78.17** | **78.87** | **80.88** | **81.81** | **82.55** |

Table 2: The comparison of different pruning methods on Tiny ImageNet. We report the Top-1 accuracy(%) of dense and pruned networks with different remaining FLOPs.

| Method | ResNet-50 (Acc: 64.28) | | | MBV3 (Acc: 63.91) | | | WRN28-10 (Acc: 61.72) | | |
|---|---|---|---|---|---|---|---|---|---|
| | 15% | 35% | 55% | 15% | 35% | 55% | 15% | 35% | 55% |
| RST-S [1] | 63.03 | 63.24 | 64.78 | 55.13 | 61.26 | 62.76 | 58.03 | 61.41 | 62.12 |
| Group-SL [14] | 0.95 | 19.94 | 55.49 | 0.56 | 2.35 | 53.43 | 0.85 | 25.74 | 57.64 |
| OTOv2 [6] | 60.38 | 63.45 | 65.16 | 57.61 | 59.25 | 60.16 | 57.19 | 61.23 | 61.70 |
| Refill [5] | 61.05 | 64.14 | 65.02 | 53.87 | 61.84 | 62.49 | 56.64 | 61.83 | 62.22 |
| Ours | **67.02** | **67.38** | **67.64** | **62.49** | **65.11** | **65.54** | **61.83** | **62.46** | **63.44** |

Table 3: Verifications of transformers on CIFAR-100. We report the Top-1 accuracy(%) of dense and pruned networks with different remaining FLOPs.

| Method | ViT (Acc: 76.49) | | | Swin (Acc: 77.16) | | |
|---|---|---|---|---|---|---|
| | 15% | 35% | 55% | 15% | 35% | 55% |
| RST-S [1] | 70.74 | 72.05 | 74.65 | 70.53 | 72.98 | 75.25 |
| Ours | **72.61** | **75.53** | **76.49** | **75.29** | **75.79** | **76.69** |

## 4.1 Benchmarking

**Results on CIFAR-100 and Tiny ImageNet**  To assess the performance of the proposed pruner and demonstrate its adaptability to various networks, we conduct experiments using CIFAR-100 and Tiny ImageNet datasets, with ResNet-50, MBV3, and WRN28-10 serving as the backbone architectures. For each dataset-network combination, we test three different FLOPs: 15%, 35%, and 55%. We compare the proposed pruner against structured RST [1] (referred to as RST-S), Group-SL [14], OTOv2 [6], and Refill [5]. All methods are evaluated under consistent training settings for a fair comparison. The results, presented in Table 1 and Table 2, reveal that the proposed pruner consistently outperforms other methods, particularly at low FLOPs. For instance, when constraint with 15% FLOPs, the proposed pruner maintains high accuracy, with gains of up to 2.73% on CIFAR-100 and 3.99% on Tiny ImageNet over the next best method.

To further validate the generalizability of the proposed pruner, we apply it to two typical Transformer models, ViT [61] and Swin Transformer [37]. Similar to the CNN experiments, we test these models on CIFAR-100 with FLOPs targets of 15%, 35%, and 55%. The results, shown in Table 3, indicate that the proposed pruner outperforms RST-S for both Transformer models across all FLOPs targets. Notably, at 55% FLOPs, the ViT pruned by the proposed method does not suffer any performance loss, and the Swin Transformer merely experiences a slight performance drop of 0.47%. The results demonstrate that while the proposed pruner is not explicitly designed for Transformers, it still achieves competitive results, highlighting its significant potential for pruning Transformer models.

**Results on ImageNet**  We further assess the performance of the proposed pruner on the prevalent ImageNet-1K benchmark. The ResNet-50 is chosen as the baseline network. Table 4 shows that, for similar FLOPs, the proposed pruner consistently suffers the least accuracy drop compared to others, underscoring the effectiveness of the proposed pruner. In the particularly challenging low FLOPs range of 10% to 20%, the proposed pruner stands out, achieving a top-1 accuracy of 73.23%, which is 3.13% higher than OTOv2, while maintaining nearly the same FLOPs (around 15%).

## 4.2 Gradient analysis

To investigate the influence of each gradient term in Algorithm 1, we conduct experiments with some of the terms disabled to observe the impact on the final performance. The results are shown in Table 5.

Table 4: Results of ResNet-50 on Imagenet. We report the Top-1 accuracy(%) of dense and pruned networks with different remaining FLOPs. The $E_{pr}$ denotes the pruning epochs. The $E_{ex}$ denotes the epochs for extra stages (such as pretraining and finetuning). The pruning epochs can be undetermined due to dynamic termination conditions, and corresponding terms are marked as "-".

| Method | Unpruned top-1 (%) | Pruned top-1 (%) | Top-1 drop (%) | FLOPs (%) | $E_{pr}$ | $E_{ex}$ |
|---|---|---|---|---|---|---|
| OTOv2 [6] | 76.10 | 70.10 | 6.00 | **14.50** | 120 | 0 |
| Refill [5] | 75.84 | 66.83 | 9.01 | 20.00 | 95 | 190 |
| **Ours** | 76.15 | **73.23** | **2.92** | 15.14 | 200 | 0 |
| MetaPruning [38] | 76.60 | 73.40 | 3.20 | **24.39** | 32 | 128 |
| Slimmable [71] | 76.10 | 72.10 | 4.00 | 26.63 | 100 | 0 |
| GAL [34] | 76.15 | 69.31 | 6.84 | 27.14 | 32 | 122 |
| DMCP [17] | 76.60 | 74.40 | 2.20 | 26.80 | 40 | 100 |
| ThiNet [45] | 72.88 | 68.42 | 4.46 | 28.50 | 110 | 90 |
| OTOv2 [6] | 76.10 | 74.30 | 1.80 | 28.70 | 120 | 0 |
| GReg-1 [62] | 76.13 | 73.75 | 2.38 | 32.68 | - | 180 |
| GReg-2 [62] | 76.13 | 73.90 | 2.23 | 32.68 | - | 180 |
| CAIE [64] | 76.13 | 72.39 | 3.74 | 32.90 | - | 120 |
| **Ours** | 76.15 | **74.43** | **1.72** | 25.31 | 200 | 0 |
| CHIP [53] | 76.15 | 75.26 | 0.89 | 37.20 | - | 270 |
| OTOv2 [6] | 76.10 | 75.20 | 0.90 | 37.30 | 120 | 0 |
| GReg-1 [62] | 76.13 | 74.85 | 1.28 | 39.06 | - | 180 |
| GReg-2 [62] | 76.13 | 74.93 | 1.20 | 39.06 | - | 180 |
| Refill [5] | 75.84 | 72.25 | 3.59 | 40.00 | 95 | 190 |
| ThiNet [45] | 72.88 | 71.01 | 1.87 | 44.17 | 110 | 90 |
| GBN [69] | 75.85 | 75.18 | 0.67 | 44.94 | 10 | 130 |
| GAL [34] | 76.15 | 71.80 | 4.35 | 45.00 | 32 | 122 |
| SCOP [59] | 76.15 | 75.26 | 0.89 | 45.40 | 140 | 90 |
| AutoPrune [65] | 74.90 | 74.50 | 0.40 | 45.46 | 60 | 90 |
| SCP [27] | 75.89 | 75.27 | 0.62 | 45.70 | 100 | 100 |
| FPGM [21] | 76.15 | 74.83 | 1.32 | 46.50 | 100 | 0 |
| LeGR [8] | 76.10 | 75.30 | 0.80 | 47.00 | - | 150 |
| AutoSlim [70] | 76.10 | 75.60 | 0.50 | 48.43 | 50 | 100 |
| AutoPruner [44] | 76.15 | 74.76 | 1.39 | 48.78 | 32 | 120 |
| MetaPruning [38] | 76.60 | 75.40 | 1.20 | 48.78 | 32 | 128 |
| CHEX [23] | 77.80 | **77.40** | 0.40 | 50.00 | 250 | 0 |
| **Ours** | 76.15 | 75.81 | **0.34** | **34.28** | 200 | 0 |
| CAIE [64] | 76.13 | 75.62 | 0.51 | 54.77 | - | 120 |
| CHIP [53] | 76.15 | 76.30 | -0.15 | 55.20 | - | 270 |
| Slimmable [71] | 76.10 | 74.90 | 1.20 | 55.69 | 100 | 0 |
| TAS [13] | 77.46 | 76.20 | 1.26 | 56.50 | 120 | 120 |
| SSS [26] | 76.12 | 71.82 | 4.30 | 56.96 | 100 | 0 |
| FPGM [21] | 76.15 | 75.59 | 0.56 | 57.80 | 100 | 0 |
| LeGR [8] | 76.10 | 75.70 | 0.40 | 58.00 | - | 150 |
| GBN [69] | 75.88 | 76.19 | -0.31 | 59.46 | 10 | 130 |
| Refill [5] | 75.84 | 74.46 | 1.38 | 60.00 | 95 | 190 |
| ThiNet [45] | 72.88 | 72.04 | 0.84 | 63.21 | 110 | 90 |
| GReg-1 [62] | 76.13 | 76.27 | -0.14 | 67.11 | - | 180 |
| MetaPruning [38] | 76.60 | 76.20 | 0.40 | 73.17 | 32 | 128 |
| **Ours** | 76.15 | **77.01** | **-0.86** | **54.38** | 200 | 0 |
| SSS [26] | 76.12 | 75.44 | 0.68 | 84.94 | 100 | 0 |
| **Ours** | 76.15 | **77.53** | **-1.38** | **76.19** | 200 | 0 |

Note that the term $g_{\mathcal{R}\rightarrow w}$ is omitted from Table 5 since it is essential to satisfy the resource constraint and is always enabled.

The addition of the term $g_{\mathcal{G}\rightarrow\theta\odot w\rightarrow\theta}$ severely degrades the accuracy by 14.22%, indicating that the gradient that aligns the soft network towards the hard one is detrimental to the final performance. Intuitively, from the perspective of parameter capacity, the hard network is practically pruned, resulting in a lower capacity than the soft network. Enforcing the soft network moving towards a less capable one is not plausible.

Both of the term $g_{\mathcal{L}\rightarrow\theta\odot w\rightarrow w}$ and $g_{\mathcal{G}\rightarrow\theta\odot w\rightarrow w}$ contribute to improve the accuracy. For the term $g_{\mathcal{L}\rightarrow\theta\odot w\rightarrow w}$, it implies searching for a mask that maximizes the performance of the soft network. The term $g_{\mathcal{G}\rightarrow\theta\odot w\rightarrow w}$ encourages the alignment of the soft and hard networks. Different from the term

Table 5: The influence of different gradient components in the proposed pruning method. The FLOPs target is set to 15% for all experiments.

| $g_{\mathcal{L}\to\theta\odot w\to\theta}$ | $g_{\mathcal{G}\to\theta\langle\hat{m}\rangle\to\theta}$ | $g_{\mathcal{G}\to\theta\odot w\to\theta}$ | $g_{\mathcal{L}\to\theta\odot w\to w}$ | $g_{\mathcal{G}\to\theta\odot w\to w}$ | Top-1 Acc (%) |
|---|---|---|---|---|---|
| ✓ | ✓ | ✓ | ✓ | ✓ | 65.55 |
| ✓ | ✓ | x | ✓ | ✓ | **79.77** |
| ✓ | x | x | ✓ | ✓ | 3.95 |
| x | ✓ | x | ✓ | ✓ | 1.73 |
| ✓ | ✓ | x | ✓ | x | 78.30 |
| ✓ | ✓ | x | x | ✓ | 78.77 |
| ✓ | ✓ | x | x | x | 77.69 |

Table 6: Gap comparison with alternative formulations of the problem 1. The symbols $\theta$, $\theta \odot w$ and $\theta\langle\hat{m}\rangle$ represent the top-1 accuracy of the original, soft and hard networks, respectively.

| Method | $JS$ | $L_2$ | Top-1 Acc (%) | | |
|---|---|---|---|---|---|
| | | | $\theta$ | $\theta\odot w$ | $\theta\langle\hat{m}\rangle$ |
| Alt 1 | 2.06e-00 | 2.74e-03 | - | - | 77.13 |
| Alt 2 | 5.17e-01 | 8.58e-04 | 78.35 | - | 77.78 |
| Ours | **1.93e-01** | **1.60e-04** | - | 80.14 | **79.77** |

$g_{\mathcal{G}\to\theta\odot w\to\theta}$, which directly imposes on massive parameters, the term $g_{\mathcal{G}\to\theta\odot w\to w}$ merely affects the learnable masks, and thus would not drastically deteriorate the soft network while improving the hard one.

The gradient term $g_{\mathcal{G}\to\theta\langle\hat{m}\rangle\to\theta}$ and $g_{\mathcal{L}\to\theta\odot w\to\theta}$ directly optimize the parameters of the hard and soft networks, respectively, leading to crucial roles in maintaining the performance. Removing either of the two terms results in an accuracy plummet of above 75%.

## 4.3 Investigation into gap

According to Section 3, we formulate the pruning problem into two parts: 1) find a superior soft network, *i.e.*, the network parameterized by $\theta \odot w$, that satisfies the resource constraint; 2) reducing the gap between the soft network and the practically pruned one, which is referred to as a hard network in this manuscript and parameterized by $\theta\langle\hat{m}\rangle$. In this section, we first provide possible alternatives to formulate the problem 1 and then compare them with our proposed one on the gap-narrowing capacity to demonstrate the superiority of our method.

The first alternative attempts to directly optimize the hard network on its performance, *i.e.*, the straight-through estimators [2]:

$$\text{Alt } 1: \min_{w}\left(\mathcal{L}\left(\theta \odot w\right)+\alpha\mathcal{R}\left(w,T\right)\right),$$
$$\min_{\theta}\mathcal{L}\left(\theta\langle\hat{m}\rangle\right). \qquad (6)$$

The second alternative substitutes the soft network with the original one while calculating the gap measure, which conforms to self-distillation-based pruners [70]:

$$\text{Alt } 2: \min_{w}\left(\mathcal{L}\left(\theta \odot w\right)+\alpha\mathcal{R}\left(w,T\right)\right),$$
$$\min_{\theta}\left(\mathcal{L}\left(\theta\right)+\mathcal{G}\left(\theta\langle\hat{m}\rangle,\theta\right)\right). \qquad (7)$$

Comparative experiments are conducted on CIFAR-100, using ResNet-50 as the baseline. The FLOPs target is set to 15%. The gap metrics, *i.e.*, the Jensen–Shannon divergence ($JS$) and $L_2$ distance, are averaged over the entire validation set. We measure the gap between the hard network and its direct supervision. For "Alt 1", the gap metrics are calculated between the 0.1 label smoothed [56] ground truth and the output of the hard network. For "Alt 2", the outputs of the original network and the hard one are utilized to calculate the gap metrics. For "Ours", the outputs of the soft network and the hard one are selected to analyze the gap.

Table 6 shows the comparison results. It can be observed that 1) a lower gap between the hard network and its direct supervision renders the hard network better performance. With the $JS$ reduced from 2.06 ("Alt 1") to 0.193 ("Ours"), the top-1 accuracy of the hard network increases from 77.13% to 79.77%; 2) Our proposed soft-to-hard formulation achieves the lowest gap on both $JS$ and $L_2$,

Table 7: The top-1 accuracy of the hard network at different fine-tuning epochs. The top-1 accuracy of the solely trained soft network before fine-tuning is 79.41%. The symbols $\boldsymbol{\theta} \odot \boldsymbol{w}$ and $\boldsymbol{\theta} \langle \hat{\boldsymbol{m}} \rangle$ represent the top-1 accuracy of the soft and hard networks, respectively.

| Epoch | | 10 | 50 | 100 | 250 | 500 |
|---|---|---|---|---|---|---|
| Top-1 Acc (%) | $\boldsymbol{\theta} \odot \boldsymbol{w}$ | 79.91 | 80.00 | 80.14 | 79.82 | 79.31 |
| | $\boldsymbol{\theta} \langle \hat{\boldsymbol{m}} \rangle$ | 76.42 | 78.89 | 79.07 | 79.49 | 79.46 |

Table 8: The top-1 accuracy of different networks pruned from ResNet-50 with a 15% FLOPs constraint and then trained from scratch without bells and whistles.

| Network | Rand 1 | Rand 2 | Rand 3 | Ours |
|---|---|---|---|---|
| Top-1 Acc (%) | 76.46 | 76.64 | 76.96 | **77.65** |

obtaining a hard network with the highest performance. The two observations imply that the soft-to-hard formulation is a relatively better scheme to narrow the gap, and the lower gap between the hard network and its direct supervision helps improve the hard network's performance.

**Can fine-tuning reduce the gap?** It might be questioned whether the coupled training of the soft and hard networks is necessary. In Section 3, we entangle the two optimizations in the problem 2 to avoid alternate optimization, which turns out to be an efficient yet effective scheme according to [36, 33]. Without the entanglement, multi-stage optimization is required. A soft network that satisfies the resource constraint is firstly trained solely, and then a fine-tuning stage attempts to narrow the gap between the soft network and the hard one. To explore the effect of fine-tuning, we train a ResNet-50 on CIFAR-100, constraint to 15% FLOPs, and merely optimize the soft network for 500 epochs. With this pretrained soft network, we perform fine-tuning via Algorithm 1 with a 0.1x learning rate and different epochs. The results can be referred to in Table 7. The fine-tuning does reduce the gap to some extent, costing 250 epochs to align the soft network and the hard one (accuracy difference drops from 3.49% to 0.33%). However, compared with our coupled training, the best accuracy of fine-tuning is still 0.28% lower at the cost of an additional 250 epochs. Consequently, the adopted coupled training turns out to be a better choice.

## 4.4 Architectural superiority

To demonstrate the architectural superiority of our pruned network, we conduct experiments on CIFAR-100, prune a ResNet-50 to 15% FLOPs via our proposed method, and then train it from scratch without bells and whistles. Three networks that are randomly pruned to 15% FLOPs are selected for the comparison. The results are shown in Table 8. The network pruned by our method achieves the highest accuracy, verifying that the pruning mask optimized via Algorithm 1 possesses architectural superiority.

## 5 Conclusion and limitations

In this paper, we reveal and study the long-standing omitted discretization gap problem in differentiable mask pruning. To bridge the discretization gap, we propose a structured differentiable mask pruning framework named Soft-to-Hard Pruner (S2HPruner), using the soft network to distill the hard network and optimize the mask. To further optimize the mask and avoid performance degradation, a decoupled bidirectional KD is proposed to alternatively maintain and block the gradient of weights and the mask. Extensive experiments verify and explain that S2HPruner can obtain high-performance hard networks with extraordinarily low resource constraints.

It is essential to acknowledge the limitations of our method. Therefore, we identify the following limitations: 1) The proposed method merely considers a single dimension, pruning feature channels of a layer. However, a block containing layers might be redundant and could be pruned as a whole, which is regarded as another pruning dimension that we do not consider in this manuscript; 2) We only validate our method on the task of image classification. It is left to explore our method's capability on other tasks, such as detection, segmentation, or natural language processing; 3) We choose FLOPs as the resource indicator, which might not ensure a hardware-friendly architecture. It is promising to consider the inference time on a specific hardware as an indicator. Above all, the identified limitations present opportunities for future research and development, and we remain committed to further exploration and refinement to overcome these challenges.

## Acknowledgement

This work is supported by National Natural Science Foundation of China (No. 62071127), National Key Research and Development Program of China (No. 2022ZD0160101), Shanghai Natural Science Foundation (No. 23ZR1402900), Shanghai Municipal Science and Technology Major Project (No.2021SHZDZX0103). The computations in this research were performed using the CFFF platform of Fudan University.

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

# Appendix A: Details of experiments

In this section, we provide the detailed specific training settings in the main manuscript. All experiments are conducted under the deep learning framework Pytorch [48], versioned 2.0.1 with Python versioned 3.10. The CUDA version is 11.8. A cluster equipped with 8 NVIDIA A100 GPUs, 1024 GB memories, and 120 CPUs is used to run experiments. A single GPU is used for experiments on CIFAR-100 and Tiny ImageNet. For Imagenet, four GPUs are paralleled to run the task.

## A1. Implementation details of CIFAR-100

The CIFAR-100 dataset [29] is a classical classification dataset, which consists of 100 categories with 50,000 training images and 10,000 testing images. For ResNet-50 [20] and MBV3 [24], we follow the training settings in [68]. In detail, the whole training epoch number is 500, and the input batch size is 64. We utilize the original SGD as the optimizer with a 0.05 initial learning rate and a 0.0003 weight decay. The cosine decay schedule is utilized to adapt the learning rate throughout the training process. For WRN28-10 [73], we follow the training settings of [73]. In detail, the epoch number and batch size are 200 and 128, respectively. The SGD is chosen as the optimizer with a 0.1 initial learning rate and a 0.0005 weight decay. The learning rate scheduler is also the cosine decay schedule. For ViT [61] and Swin Transformer [37], we use an image size of 32x32 and a patch size of 4. The epoch number and batch size are 200 and 128, respectively. The optimizer is AdamW [42] with an initial learning rate of 0.001/0.003 for Swin/ViT and a 0.05 weight decay. The learning rate is warmed up for 10 epochs. The data augmentations are the same as the ones in [31]. Different from CNNs, where we regard the channel numbers of convolutional and linear layers as the width dimension, to prune the width of Transformers, we take the head numbers (ViT) or head feature dimensions (Swin) of attention layers and the channel numbers of linear layers into account.

## A2. Implementation details of Tiny ImageNet

The Tiny ImageNet dataset is derived from the renowned ImageNet dataset [11], comprising 200 categories, 100,000 training images, and 10,000 test images. For the ResNet-50 [20] and MBV3 [24] models, we employ 500 epochs and a batch size of 64. The optimization is performed using SGD with an initial learning rate of 0.1 and a weight decay of 0.0003. We utilize a step-wise learning rate scheduler, reducing the learning rate to 0.1 and 0.01 of the original at the 250th and 375th epochs, respectively. For the WRN28-10 [73] architecture, we adopt the training settings from [51], with 200 epochs and a batch size of 128. The SGD optimizer is used with an initial learning rate of 0.2 and a weight decay of 0.0001. The learning rate is decreased in a step-wise manner, dropping to 0.1 and 0.01 of the initial value at the 100th and 150th epochs, respectively.

## A3. Implementation details of ImageNet

The ImageNet dataset [11] is a widely used classification benchmark, containing 1,000 categories, 1.2 million training images, and 50,000 testing images. For the evaluated ResNet-50 [20], the epoch number and batch size are 200 and 512, respectively. We utilize SGD as the optimizer. The learning rate is initialized as 0.2 and is controlled by a cosine decay schedule. The weight decay is 0.0001. Besides, we apply the commonly used data augmentations according to [25, 55].

## A4. Hyperparameters $\alpha$, $\beta$, and $\gamma$

To determine the hyperparameters in Algorithm 1, we utilize a dynamic balancing scheme based on the $L_2$ norm of gradients. Specifically, the $g_{\mathcal{L} \to \theta \odot w \to w}$ and $g_{\mathcal{G} \to \theta \odot w \to w}$ are firstly normalized by their own $L_2$ norms before being added together. The addition result is then aligned with $g_{\mathcal{R} \to w}$ via being scaled to the $L_2$ norm of $g_{\mathcal{R} \to w}$. For $g_{\mathcal{L} \to \theta \odot w \to \theta}$ and $g_{\mathcal{G} \to \theta \langle \hat{m} \rangle \to \theta}$, no balancing is applied. The two terms are added with fixed coefficients. For CNNs, the coefficients are 0.5 and 5 for $g_{\mathcal{L} \to \theta \odot w \to \theta}$ and $g_{\mathcal{G} \to \theta \langle \hat{m} \rangle \to \theta}$, respectively. For Transformers, the coefficients are 1 and 1 for $g_{\mathcal{L} \to \theta \odot w \to \theta}$ and $g_{\mathcal{G} \to \theta \langle \hat{m} \rangle \to \theta}$, respectively. The coefficient for $g_{\mathcal{R} \to w}$ is set to 5.

## Appendix B: Trajectory of FLOPs and accuracy

In this section, the FLOPs and accuracy trajectory is provided to display the pruning procedure of S2HPruner visually. We conduct experiments on five different models, including ResNet-50 [20], MobileNetV3 (MBV3) [24], WideResNet28-10 [73], ViT [61], and Swin Transformer [37] on CIFAR-100. The results are shown in Fig. 3 and as the training epoch increases, our methods can fast converge the capacity of the hard network to the target FLOPs. However, it does not mean the mask optimization is finished. It can be seen that the performance of the robust network is steadily improving. It suggests that after entering the feasible region, S2HPruner consistently explores the possible structure and exploits the optimal architecture. Moreover, although applied to five unique architectures, S2HPruner obtains similar trajectories, which demonstrates the generalization of S2HPruner.

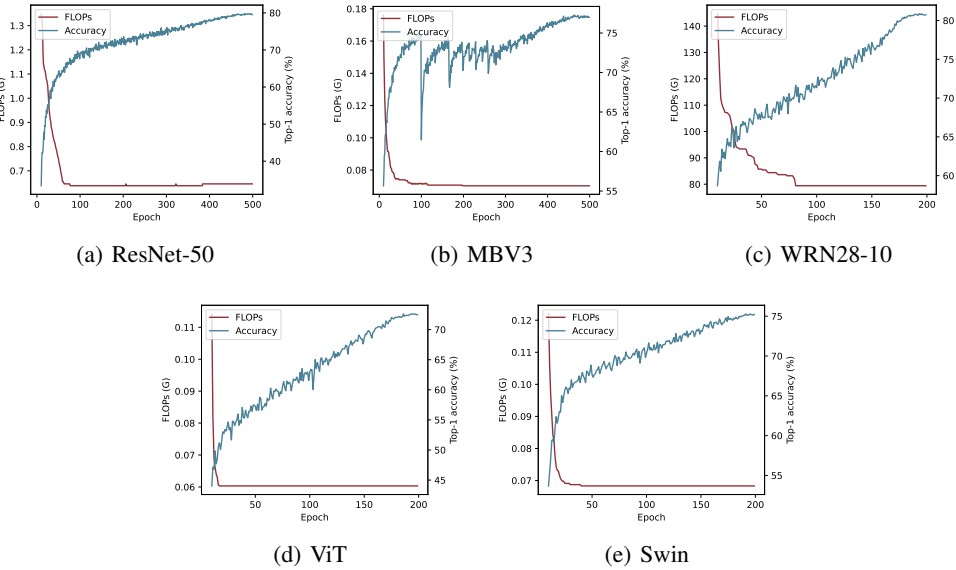

Figure 3: The trajectory of FLOPs and accuracy. We report the accuracy and FLOPs of the hard network during the training of different models, including (a) ResNet-50 (b) MobileNetV3 (c) WideResNet28-10 (d) ViT (e) Swin Transformer on CIFAR-100.

## Appendix C: Visualization of pruning process

We report the detailed output channel variation of different five networks during pruning visually. The results are shown in Fig 5, 6, 7, 8, 9. The target FLOPs is set to 15%. It is worth noting that because the mask is dependent on the dependencies groups where layers all have the same output channels, we report the index of dependencies groups as the index of layers, which does not correspond to the raw definition completely. It can be observed that the channel variation is disparate between different layers, which implies our method is not restricted to trivial solutions such as uniform channel distribution. Combined analysis with Fig. 3, we can observe that although the FLOPs satisfies the constraints, our method is not caught in loafing but can consistently explore the structure space to find the optimal architecture. A similar phenomenon also exists in all five networks, which demonstrates the generalization of the proposed method.

## Appendix D: The architecture of the pruned network

We provide the architecures of our pruned networks in Fig. 4. The pruned networks are obtained via using Algorithm 1 on CIFAR-100 with a 15% FLOPs target. It can be observed from Fig. 4 that different pruned network varies in architecture pattern. For example, convolutional neural networks (CNNs), *i.e.*, ResNet-50, MBV3, and WRN28-10 may prefer deeper layers. The retained channels are

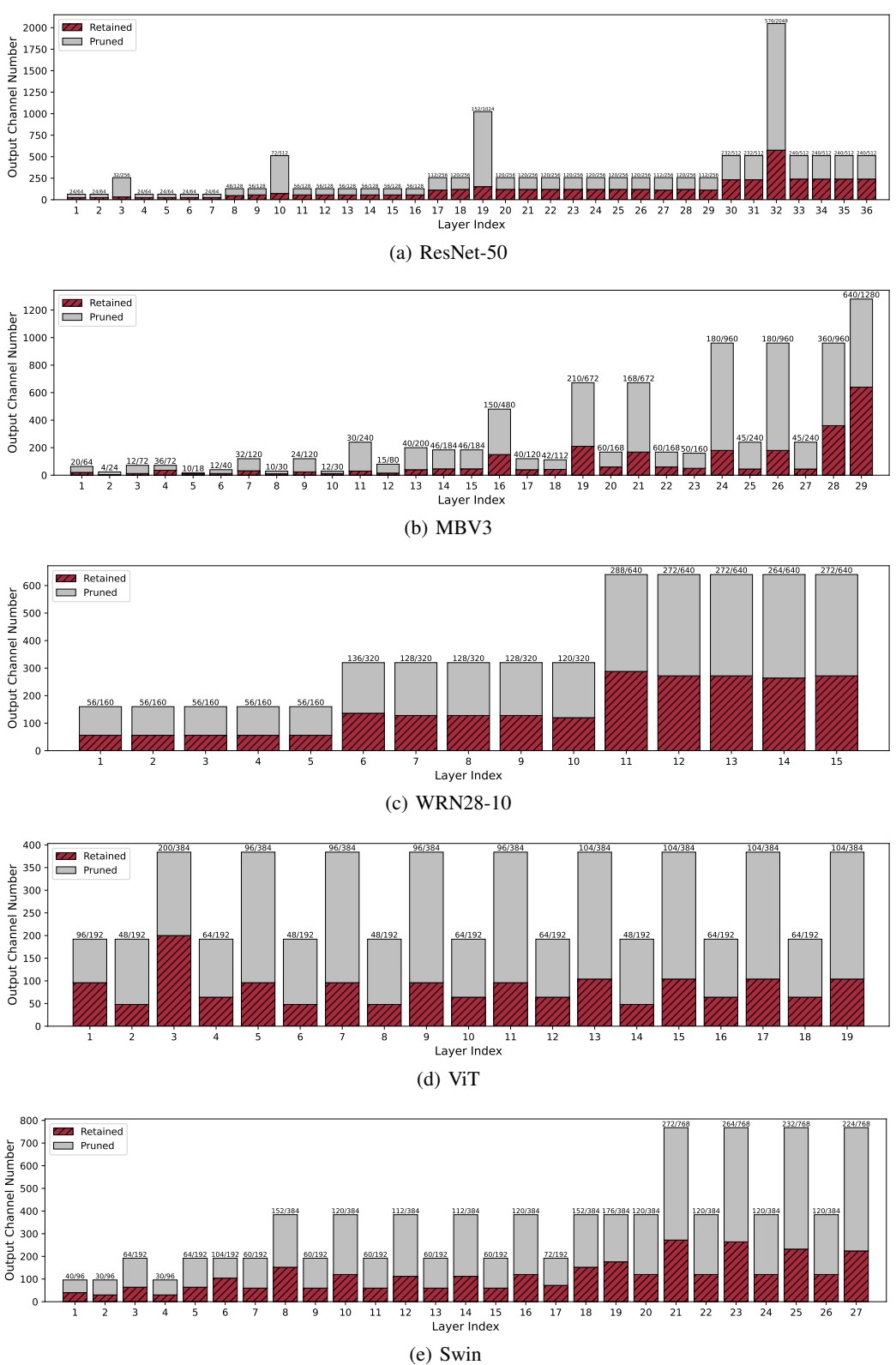

Figure 4: The architectures of networks, including (a) ResNet-50 (b) MobileNetV3 (c) WideResNet28-10 (d) ViT (e) Swin Transformer, pruned via our proposed method on CIFAR-100. The target FLOPs is set to 15%.

concentratively distributed in the post-half layers. Different from CNNs, Transformers, *i.e.*, ViT, and Swin seem not to exhibit an obvious preference for layer depth. The pruning pattern of the shallow layers is almost uniform with that of the deep layers.

Table 9: The pruning results obtained via training a ResNet-50 on CIFAR-100 with different random seeds using our proposed method. We report the Top-1 accuracy and FLOPs.

| Exp | #1 | #2 | #3 | #4 |
|---|---|---|---|---|
| Top-1 Acc (%) | 79.77 | 79.68 | 79.80 | 79.75 |
| FLOPs (%) | 15.36 | 15.22 | 15.94 | 15.21 |

Table 10: Training efficiency comparison with different methods. For a fair comparison, double-epoch training results of other methods are included.

| | RST-S | Depgraph | OTO v2 | IMP-Refill | Ours |
|---|---|---|---|---|---|
| Top-1 Acc (%) (1x training schedule) | 75.02 | 49.07 | 77.04 | 75.12 | 79.77 |
| Top-1 Acc (%) (2x training schedule) | 75.54 | 50.83 | 77.21 | 75.66 | - |
| GPU time per epoch (s) | 44.50 | 70.97 | 79.36 | 74.12 | 50.13 |
| Peak GPU memory (MB) (training) | 4329 | 4319 | 4221 | 4261 | 4710 |
| Peak GPU memory (MB) (inference) | 1351 | 1365 | 1262 | 1329 | 1279 |

## Appendix E: Robustness against randomness

To assess the consistency of our proposed pruning method, we target a 15% reduction in FLOPs using ResNet-50 as the base model on the CIFAR-100 dataset. Four independent runs with varying random seeds are conducted, and the results are presented in Table 9. The pruned networks consistently achieved comparable performance, with negligible variations in Top-1 accuracy (less than 0.1% deviation) and FLOPs (less than 1% deviation). These findings validate the robustness of our proposed method, indicating that the resource consumption of the pruned network is expected and its performance is reliable.

## Appendix F: Training efficiency

To investigate the training efficiency of the proposed method, we compare its training time to other established pruning methods in Table 10. Using ResNet-50 on the CIFAR-100 dataset, our experiments reveal that the proposed method achieves exceptional performance while maintaining a competitive training time, ranking second-shortest among the tested methods. This efficiency stems from the inherent parallelism of the soft and hard networks. The forward and backward passes of the soft and hard networks can be executed simultaneously, leveraging the power of CUDA streams or multi-GPU parallelism. Furthermore, our method operates in a single stage, eliminating the need for sequential fine-tuning or iterative pruning, further contributing to its time efficiency. To isolate the impact of forward/backward pass counts, we extended the training epochs of other methods two-fold to match our method's counts. Despite this, the performance of these methods plateaued, indicating that simply increasing training time does not guarantee improved pruning results. This underscores the inherent advantages of our method.

Besides, the GPU memory costs during training and inference are also reported in Table 10. During training, our method costs bearable (about 10%) more GPU memories than the average of other methods due to the additional learnable masks and the mask state buffers in the optimizer. During inference, the GPU memory costs merely depend on the scale of the pruned network. As the FLOPs target is set to 15% for all the methods, there is no significant difference in GPU memory costs.

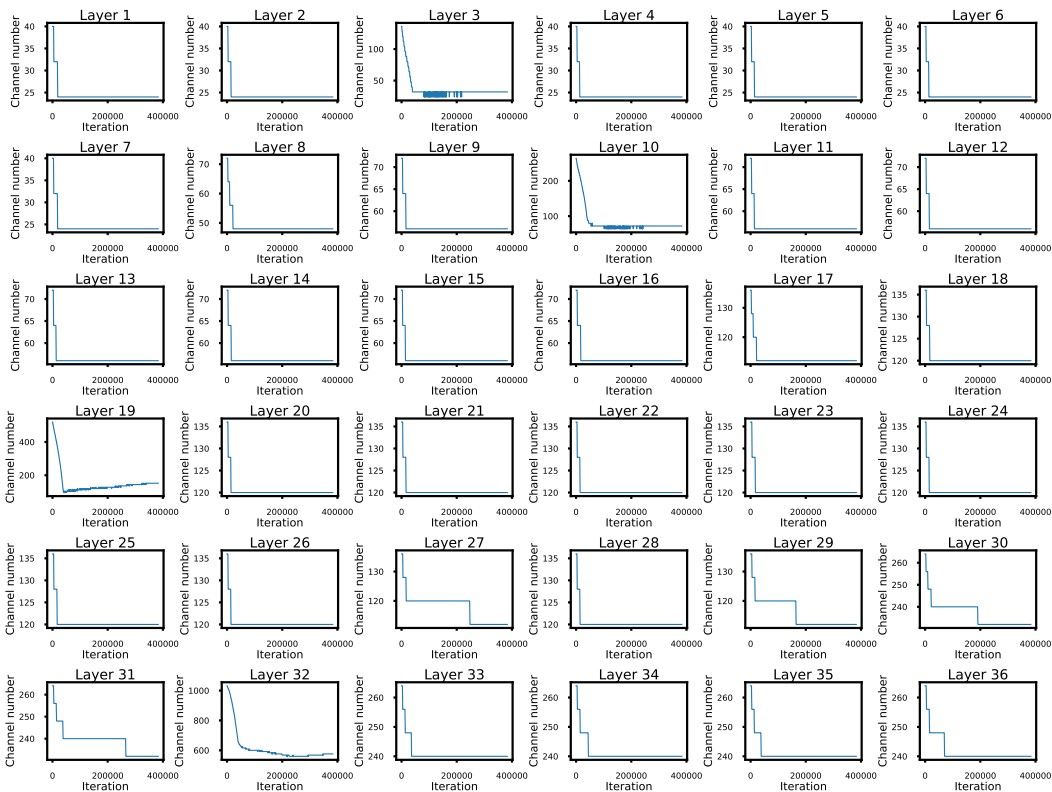

Figure 5: The detailed channel variation of ResNet-50 on CIFAR-100 during training. The target FLOPs is set to 15%. The horizontal axis represents the training iterations. The vertical axis represents the output channel number.

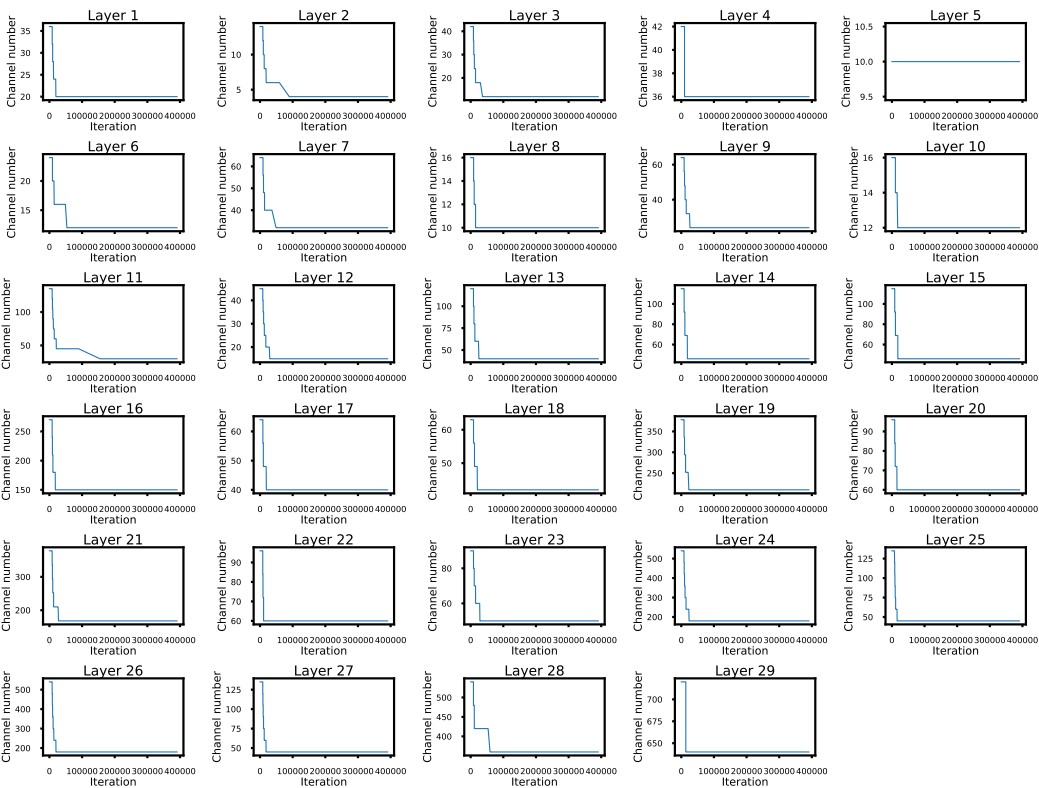

Figure 6: The detailed channel variation of MobileNetV3 on CIFAR-100 during training. The target FLOPs is set to 15%. The horizontal axis represents the training iterations. The vertical axis represents the output channel number.

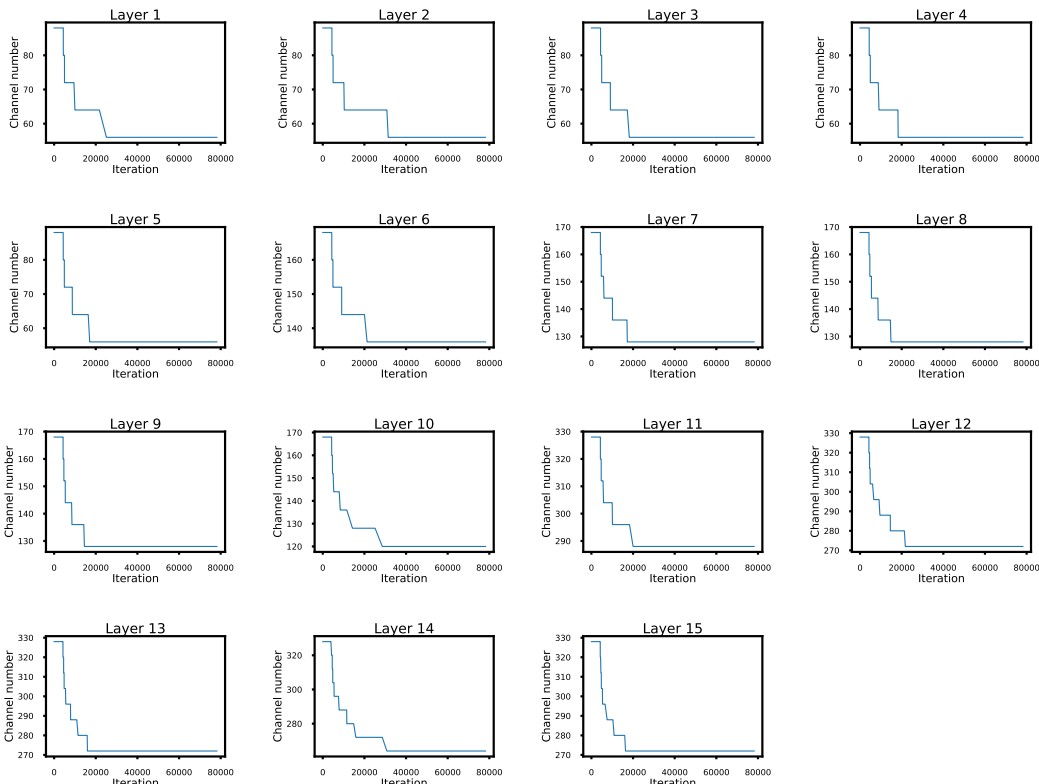

Figure 7: The detailed channel variation of WideResNet28-10 on CIFAR-100 during training. The target FLOPs is set to 15%. The horizontal axis represents the training iterations. The vertical axis represents the output channel number.

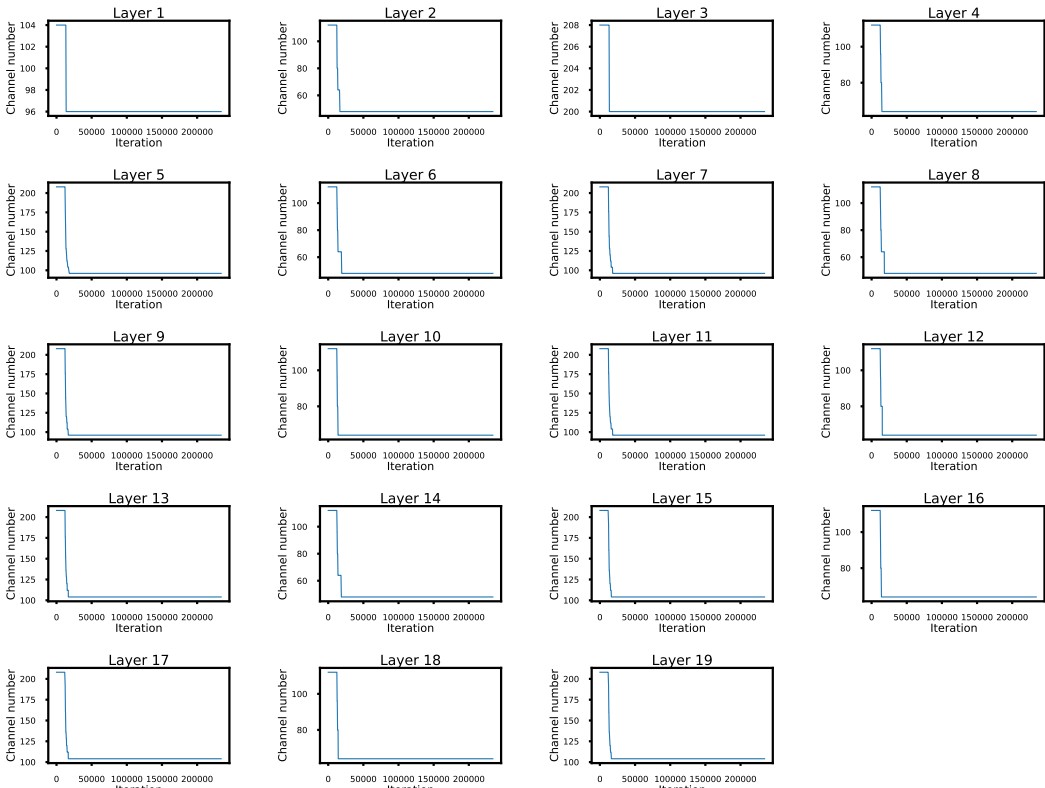

Figure 8: The detailed channel variation of ViT on CIFAR-100 during training. The target FLOPs is set to 15%. The horizontal axis represents the training iterations. The vertical axis represents the output channel number.

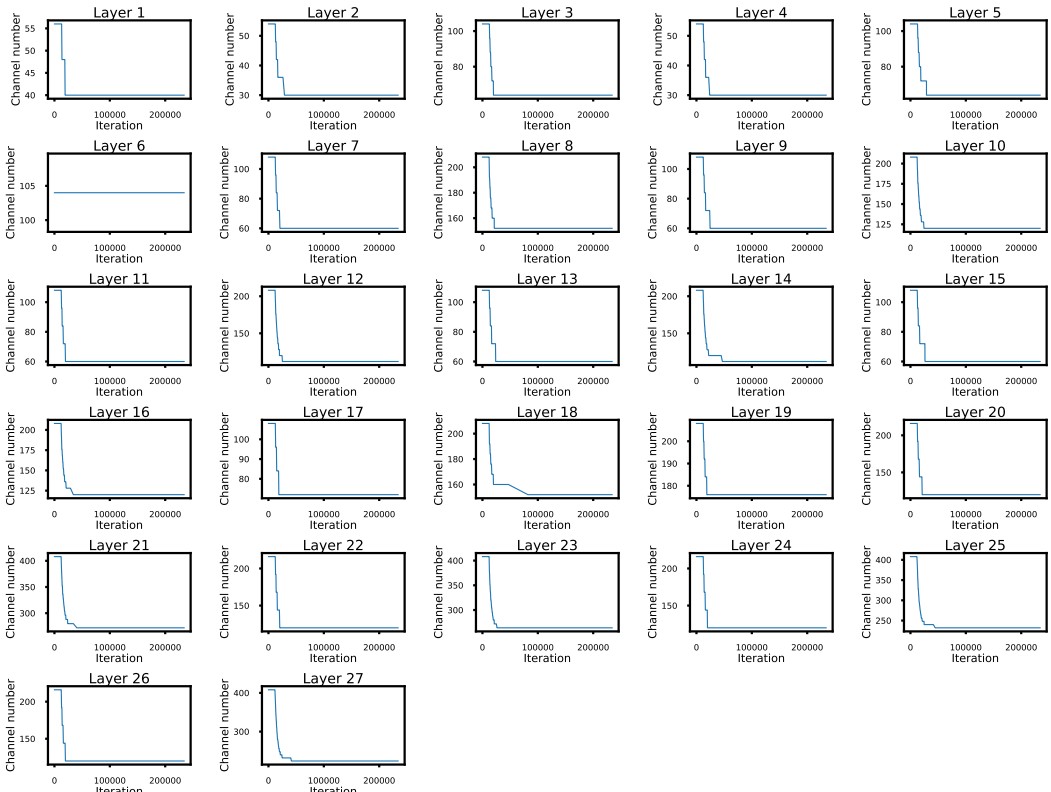

Figure 9: The detailed channel variation of Swin Transformer on CIFAR-100 during training. The target FLOPs is set to 15%. The horizontal axis represents the training iterations. The vertical axis represents the output channel number.

