# OpenReview forum: "S2HPruner: Soft-to-Hard Distillation Bridges the Discretization Gap in Pruning"
_NeurIPS.cc/2024/Conference — NeurIPS 2024 poster_

### Official Review · Reviewer_ECuu · 2024-07-10

**Soundness:** 3
**Presentation:** 3
**Contribution:** 3
**Rating:** 6
**Confidence:** 5

**Summary:**

The paper titled "S2HPruner: Soft-to-Hard Distillation Bridges the Discretization Gap in Pruning" introduces a novel framework designed to address the challenges associated with the discretization gap in neural network pruning techniques. The authors propose a method that bridges the gap between the representation capacities of a continuously relaxed network (soft network) and its pruned counterpart (hard network), aiming to improve the pruning performance without requiring post-pruning fine-tuning.

**Strengths:**

1. The S2HPruner framework is innovative in that it directly addresses the issue of the discretization gap, which is often overlooked. By incorporating soft-to-hard distillation, the method ensures that the hard network maintains a similar level of performance to the soft network, which is a significant improvement over traditional pruning methods.
2. The overall writing is good and the work is easy to follow.

**Weaknesses:**

1. The figures can be further improved in this work. For instance, figure 2 is not good engouth to illustrated the overall pipeline of the work. It is recommanded to explictly show soft net and hard net instead of thier outputs only.
2. The formats of the tables are not consistent.

**Questions:**

See weaknesses above

**Limitations:**

It is recommanded to discuss the training time and GPU RAM cost during training and inference. The comparisons between the proposed one and existing SOTAs are necessary.

---

> ### Author Rebuttal · Authors · 2024-08-05
>
> **Q1: Figure 2 is recommended to explicitly show soft and hard networks.**
>
> *Answer:* Thanks for providing the suggestion. We will emphasize the concept of "soft" and "hard" in Figure 2 and try our best to make it more comprehensible. The revised version can be referred to in Figure t2.
>
> &nbsp;
>
> **Q2: The formats of the tables are not consistent.**
>
> *Answer:* Following the advice, we will check all the tables and try our best to align their formats in the revised manuscript.
>
> &nbsp;
>
> **Q3: It is recommended to discuss the training time, and GPU RAM cost during training and inference.**
>
> *Answer:* The discussion about training time can be found in Appendix F, where we compare our method with four SOTA methods that we can reproduce. To further discuss GPU memory costs, we expand the table in Appendix F with additional metrics, as shown in Table t4. During training, our method costs bearable (about 10\%) more GPU memories than the average of other methods due to the additional learnable masks and the mask state buffers in the optimizer. During inference, the GPU memory costs merely depend on the scale of the pruned network. As the FLOPs target is set to 15\% for all the methods, there is no significant difference in GPU memory costs. Detailed discussion about the efficiency of our method can be found in Appendix F.

---

### Official Review · Reviewer_aPhZ · 2024-07-12

**Soundness:** 3
**Presentation:** 3
**Contribution:** 3
**Rating:** 5
**Confidence:** 3

**Summary:**

In this article, the author proposes using a 0-1 mask (hard network) and a differentiable mask (soft network) with an accuracy gap as a starting point for network distillation, where the distillation function selects Kullback Leibler divergence as the gap measure（S2HPruner）. This method was tested on datasets including CIFAR-100, Tiny ImageNet, and ImageNet for Resnet-50, ViT Swin, and other models.

**Strengths:**

* This paper is easy to follow.
* The experiments and investigation into gaps and gradients are well-developed.

**Weaknesses:**

**Q1:**  The abstract section mentions the concept of 'bidirectional', but it's incorrect. My understanding is that S2HPruner only uses sparsity in forward flows and does not involve sparsity in backward flows. Please explain the meaning of this "bidirectional" in detail.

**Q2:** Is there a reason to utilize Kullback Leibler divergence when selecting a gap measure? Is it possible to add corresponding experiments or proof explanations to the article that does not provide an explanation?

**Q3:** How does S2HPruner control network's sparsity? The method section of the article lacked any description. From the description in Figure 2, it appears that the mean is calculated from a differentiable soft mask as a threshold, and if this is the case, it seems that the purpose of controlling the sparsity rate will not be achieved.

**Questions:**

Please see the weakness part

**Limitations:**

yes

---

> ### Author Rebuttal · Authors · 2024-08-05
>
> **Q1: Explain the meaning of "bidirectional" in detail.**
>
> *Answer:* We use "bidirectional" to describe that the knowledge transfer in our method is bidirectional. The soft network transfers knowledge to the hard one, improving the performance of the hard network. Simultaneously, the hard network provides guidance for the soft network to better optimize the mask. Both directions of knowledge transfer can benefit the reduction of the discretization gap in pruning. Detailed explanations can be referred to in lines 59-63 in the main manuscript.
>
> &nbsp;
>
> **Q2: Why is Kullback-Leibler divergence selected as the gap measure?**
>
> *Answer:* The Kullback-Leibler divergence has been proved advantageous to reduce the gap between two distributions [1,2]. To verify its superiority, we keep other training settings unaltered and compare the Kullback-Leibler divergence with two well-known metrics, L1 and L2 distance. The results are reported in Table t3. The Kullback-Leibler divergence exhibits a distinct advantage over the other two metrics, turning out that it is suitable to be selected as a gap measure.
>
> &nbsp;
>
> **Q3: How does the network control sparsity?**
>
> *Answer:* To control sparsity, we introduce a resource regularization $\mathcal{R}=\left(FP_{soft}/FP_{all}-T\right)^2$ as stated in lines 153-157 in the main manuscript. Given a FLOPs target, the resource regularization controls the soft output channel number of each layer in a differentiable manner to gradually meet the target. In line 157, the soft output channel number is presented as $C_s=\sum_{k=1}^{C_{out}}\left(u_{k}*k\right)$. Note that, the "differentiable soft mask" mentioned in the question is the $w_i$, derived from $\sum_{k=i}^{C_{out}}u_k$ according to line 151. As a result, the mean of the $w_i$ can be derived as $C_{s}/C_{out}$. After thresholding, the hard output channel number is $C_h=\left|\left\\{j\in\left[1,C_{out}\right]\left|w_j\geq\left(C_{s}/C_{out}\right)\right.\right\\}\right|$. According to [3], the $C_h$ is close to $C_s$ in most cases and can be finally converged to satisfy the FLOPs target of the hard network. Moreover, we provide the difference of $C_s$ and $C_h$ during training in Figure t1. It can be observed that the difference is negligible all over the training process.
>
> &nbsp;
>
> [1]: Hinton, G., Vinyals, O., Dean, J.: Distilling the knowledge in a neural network. arXiv preprint arXiv:1503.02531 (2015)
>
> [2]: Tang, S., Ye, P., Li, B., Lin, W., Chen, T., He, T., Yu, C., Ouyang, W.: Boosting residual networks with group knowledge. In: Proceedings of the AAAI Conference on Artificial Intelligence. vol. 38, pp. 5162–5170 (2024)
>
> [3]: Chen, M., Shao, W., Xu, P., Lin, M., Zhang, K., Chao, F., Ji, R., Qiao, Y., Luo, P.: Diffrate: Differentiable compression rate for efficient vision transformers. In: Proceedings of the IEEE/CVF International Conference on Computer Vision. pp.138 17164–17174 (2023)

---

### Official Review · Reviewer_MZh4 · 2024-07-28

**Soundness:** 3
**Presentation:** 3
**Contribution:** 3
**Rating:** 6
**Confidence:** 5

**Summary:**

Discretization in pruning poses a huge threat to network performance. To alleviate this issue, the paper proposes S2HPruner, a pruning method that leverages distillation. In details, the pruning process involves two networks that share the same architecture. The difference is that the teacher network has a weight covered with soft, differentiable masks, while the pruned student network weight is covered with non-differentiable binary masks. The optimization target involves a distillation loss that narrows the gap between the differentiable and non-differentiable network. The proposed method achieves good performance on various datasets.

**Strengths:**

The paper highlights the harm of discretization in pruning, which I think is the core problem.
The paper distill knowledge from a soft-mask network to a hard-mask one, which I think is novel.

**Weaknesses:**

1. Line 9 in Abstract: SH2->S2H

2. Notation problem: the authors should unify the gradient notations in the pseudocode and the equations.

3. For Table 4: as different methods use different training settings, the number of the training epochs should be indicated.

4. The baselines in Table 4 are a bit out-of-date. The proposed S2HPruner should be compared against latest channel pruning methods, like SCOP [1] and CHEX [2].

[1] github.com/yehuitang/Pruning/tree/master/SCOP_NeurIPS2020

[2] github.com/zejiangh/Filter-GaP

**Questions:**

1. Baselines reflect the actual training capability of a certain codebase. Why do you use 76.15 instead of 76.8 as the ResNet-50 baseline? Please justify.

2. From a high-level perspective, the method is a combination of distillation and pruning. I wonder if the method could outperform the simple combination of pruning and knowledge distillation (at equal training costs)?

**Limitations:**

Limitations are addressed.

---

> ### Author Rebuttal · Authors · 2024-08-05
>
> **Q1: The typos in Abstract and the inconsistency of gradient notations in the pseudocode and the equations.**
>
> *Answer:* Thanks for the suggestion. We will fix the typos and align the notation of gradients in the pseudo code and the equations.
>
> &nbsp;
>
> **Q2: In Table 4, the epochs should be indicated and additional baselines, SCOP and CHEX, are required.**
>
> *Answer:* Following the advice, we will report the epochs required to obtain a pruned network from scratch in Table 4 like Table t1. For methods pruning from a pretrained model, we report the pretraining and pruning epochs, respectively. In Table 4, we aim to select widely used pruning methods with a large range of time for comprehensive comparison, and Table 4 also includes some of the latest methods such as OTOv2 (2023) [2] and Refill (2022) [4]. Following R1's suggestions, we also compare S2HPruner with the mentioned baselines SCOP [1] and CHEX [5]. Specifically, for SCOP, because it has a similar unpruned top-1 accuracy as ours, we directly report the results from its original literature. For CHEX, its unpruned top-1 accuracy is significantly larger than ours. Thus, we additionally conduct an experiment that deploys the standard CHEX on our training schedule, named CHEX*. The results are shown in Table t1. It can be observed that, under different FLOPs constraints, our method consistently suffers the minimal Top-1 drop, demonstrating the superiority of our method. Moreover, under the same training schedule, S2HPruner can outperform CHEX with higher accuracy and lower FLOPs. The above experiments and comparisons will be included in Table 4 in the revision.
>
> &nbsp;
>
> **Q3: Why is the baseline of ResNet-50 76.15\%?**
>
> *Answer:* The 76.15\% Top-1 accuracy is obtained via training a ResNet-50 baseline on ImageNet with the recipe of our codebase, which is reported in Appendix A3. Besides, it is hard to compare different pruning methods fairly due to their distinct training schedule, and the Top-1 accuracy around 76.15\% is a mainstream and typical benchmark adopted by recent pruning works, such as the mentioned SCOP [1], OTOv2 [2], and Greg-2 [3]. Considering it, in this manuscript, we adopt the ResNet-50 with 76.15\% Top-1 accuracy on ImageNet as the dense model for pruning for fair comparison among different pruning methods.
>
> &nbsp;
>
> **Q4: Whether the proposed method could outperform the simple combination of pruning and knowledge distillation (at equal training costs).**
>
> *Answer:* To compare with the simple combination of pruning and knowledge distillation at equal training costs, we carry out experiments using ResNet-50 on CIFAR-100. Firstly, an STE pruner (see Eq. 6 in the main manuscript) is utilized to prune a network under 15\% FLOPs constraint for half of the total epochs reported in Appendix A1. Then, vanilla knowledge distillation is applied for the remaining half of the total epochs. The teacher network is a dense ResNet-50 with full training on CIFAR-100, and the student network is the network pruned by STE pruner. The results are shown in Table t2. Our method outperforms the simple combination in Top-1 accuracy by 2.13\%. It demonstrates that knowledge distillation is tightly integrated into our pruning process, and the joint optimization renders the pruned network better performance. Detailed statements can be found in lines 103-108 in the main manuscript.
>
> &nbsp;
>
> [1]: Tang, Y., Wang, Y., Xu, Y., Tao, D., Xu, C., Xu, C., Xu, C.: Scop: Scientific control for reliable neural network pruning. Advances in Neural Information Processing Systems 33 160 (2020)
>
> [2]: Chen, T., Liang, L., Tianyu, D., Zhu, Z., Zharkov, I.: Otov2: Automatic, generic, user-friendly. In: International Conference on Learning Representations (2023)
>
> [3]: Wang, H., Qin, C., Zhang, Y., Fu, Y.: Neural pruning via growing regularization. In: International Conference on Learning Representations (ICLR) (2021)
>
> [4]: Chen, T., Chen, X., Ma, X., Wang, Y., Wang, Z.: Coarsening the granularity: Towards structurally sparse lottery tickets. In: International Conference on Machine Learning. pp.142 3025–3039. PMLR (2022)
>
> [5]: Hou, Z., Qin, M., Sun, F., Ma, X., Yuan, K., Xu, Y., Chen, Y.K., Jin, R., Xie, Y., Kung, S.Y.: Chex: Channel exploration for cnn model compression. In: Proceedings of the IEEE/CVF Conference on Computer Vision and Pattern Recognition. pp.152 12287–12298 (2022)

---

> > ### Comment · Reviewer_MZh4 · 2024-08-11
> > **Response to Rebuttal**
> >
> > I am satisfied with the results presented by the author, and I will raise my score. I also hope that the author could include these results in the paper upon acceptance.

---

> > > ### Author Response · Authors · 2024-08-11
> > >
> > > Thanks for your approval! We promise to include these results in the revision and cite the corresponding methods.

---

### Author Rebuttal · Authors · 2024-08-05

Thanks for the valuable feedback provided by all reviewers. We appreciate the reviewers MZh4 (R1), aPhZ (R2), and ECuu (R3) for approving our contributions: (1)  innovative method (R1, R3), (2) well-developed experiments (R2), (3) good writing and easy to follow (R2, R3). Besides, the concerns are mainly concentrated on (1) format problems (R1, R3), (2) the figure of the framework (R3), (3) some declaration on settings and definitions (R1, R2, R3). Under the NeurIPS policy, we will follow reviewers’ suggestions to refine the figures and format of the paper at our discretion. Below, please find our detailed answers to address your concerns. All tables and figures tagged by "tx" are represented in the appended pdf file.

---

### Decision · Program_Chairs · 2024-09-25

**Decision:**

Accept (poster)

**Comment:**

This paper introduces a novel framework to address the challenges associated with the discretization gap in neural network pruning techniques. The reviewers identify the novelty, the experimental performance, and the writing of the paper and point out several minor issues at the same time. In rebuttal, the authors carefully addressed the issues mentioned by the reviewers and the current version looks ready for publication.